# The Role of DNA/Histone Modifying Enzymes and Chromatin Remodeling Complexes in Testicular Germ Cell Tumors

**DOI:** 10.3390/cancers11010006

**Published:** 2018-12-20

**Authors:** João Lobo, Rui Henrique, Carmen Jerónimo

**Affiliations:** 1Cancer Biology and Epigenetics Group, Research Center of Portuguese Oncology Institute of Porto (GEBC CI-IPOP), R. Dr. António Bernardino de Almeida, 4200-072 Porto, Portugal; joaomachadolobo@gmail.com (J.L.); henrique@ipoporto.min-saude.pt (R.H.); 2Department of Pathology, Portuguese Oncology Institute of Porto (IPOP), R. Dr. António Bernardino de Almeida, 4200-072 Porto, Portugal; 3Department of Pathology and Molecular Immunology, Institute of Biomedical Sciences Abel Salazar, University of Porto (ICBAS-UP), Rua Jorge Viterbo Ferreira 228, 4050-513 Porto, Portugal

**Keywords:** testicular germ-cell tumors, histone modifications, chromatin remodeling, methylation, epigenetics, biomarkers

## Abstract

It is well established that cancer cells exhibit alterations in chromatin structure and accessibility. Indeed, the dysregulation of many protein-coding players with enzymatic activity (DNA and histone-modifying enzymes) and chromatin remodelers have been depicted in various tumor models in recent years. Still, little attention has been directed towards testicular germ cell tumors (TGCTs)—representing the most common neoplasm among young adult Caucasian men—with most studies focusing on exploring the role of DNA methyltransferases (*DNMTs*) and DNA demethylases (*TETs*). TGCTs represent a complex tumor model, associated with developmental and embryogenesis-related phenomena, and display seldom (cyto)genetic aberrations, leaving room for Epigenetics to explain such morphological and clinical diversity. Herein, we have summarized the major findings that were reported in literature regarding the dysregulation of DNA/histone-modifying enzymes and chromatin remodelers in TGCTs. Additionally, we performed in silico analysis of The Cancer Genome Atlas database to find the most relevant of those players in TGCTs. We concluded that several DNA/histone-modifying enzymes and chromatin remodelers may serve as biomarkers for subtyping, dictating prognosis and survival, and, possibly, for serving as targets of directed, less toxic therapies.

## 1. Testicular Germ Cell Tumors in Brief

Testicular germ cell tumors (TGCTs) comprise more than 95% of testicular neoplasms and they are grouped in two major families according to the most recent World Health Organization classification: the germ-cell neoplasia in situ (GCNIS)-related tumors (the most frequent, which include Seminomas (SEs) and Non-Seminomatous Tumors (NSTs), two subgroups with very distinct behavior and clinical impact), and the GCNIS-unrelated ones [1,2].

Despite representing only 1% of male cancer worldwide, they constitute the most common cancer afflicting Caucasian men between 15–44 years old, with the Western lifestyle contributing to a rising incidence. They also exhibit outstanding cure rates and a decreasing mortality trend, in response to multimodal treatments. However, many issues are left unresolved and they deserve our attention, namely the substantial proportion of patients with disseminated disease that relapse with poor prognosis, the emergence of cisplatin resistance, and the considerable morbidity induced by chemo- and radiotherapy in such young patients with long lifetime expectancy [3,4].

TGCTs are remarkably heterogeneous (reflecting the complexity of this tumor model) but they mainly share a unifying cytogenetic background and display very few mutations. In this line, it is only natural that various Epi-phenomena might play a fundamental role in these neoplasms. Therefore, the study of new Epi-markers might aid in tumor subtype discrimination, prognosis assessment, and disease monitoring, as no accurate validated biomarkers exist for these purposes. Also, the manipulation of these Epi-markers might provide ways of uncovering therapies with improved antitumor activity, less toxicity, and that may overcome cisplatin resistance [5,6,7,8,9,10,11].

## 2. Protein-Coding Epigenetic Players: Their Role in Cancer

Histones and non-histone proteins undergo post-translational modifications (PTMs), the most studied being methylation, acetylation, and phosphorylation, which alter the chromatin pattern, hence controlling gene expression. Also, chromatin remodeling complexes (ChRC) alter the nucleosome structure, with implications in DNA accessibility. Another type of proteins (methyltransferases and demethylases) are responsible for regulating (i.e., writing and erasing) DNA methylation, again with implications in gene expression and chromatin stability. In this line, all of these players are involved in fundamental biological processes, such as cell division and proliferation, cell cycle, metabolism, pluripotency, genomic imprinting, and DNA repair, and globally regulate the transcription of many genes. Therefore, it is only rational to think that these epigenetic mechanisms are deregulated in cancer, and that they can be modulated to treat these patients [12,13,14,15,16]. In fact, cancer cells globally display hypomethylation (contributing to genomic instability) along with a preferential hyper/hypomethylation of promoter-associated CpG islands of tumor suppressor genes and oncogenes, respectively. Histone-modifying enzymes and ChRCs cooperate in modulating gene expression profile, upregulating oncogenes, and downregulating tumor suppressors. All in all, these players have been shown to be relevant to all the steps of tumorigenesis, in various models [14,17,18,19,20,21,22,23,24,25].

## 3. Protein-Coding Epigenetic Players in Testicular Germ Cell Tumors

The field of Epigenetics in TGCTs has been expanding in the last years, with a growing number of publications on the topic. Most studies have focused on methylation [26,27,28,29] and on microRNAs (miRs) [30,31,32,33], where the major breakthroughs in TGCTs have taken place. Protein-coding epigenetic players, including DNA-modifying enzymes, histone-modifying enzymes, and ChRCs, have been explored in diverse tumor models in the recent years; however, little attention has been paid to TGCTs. Hence, we have conducted a PubMed search with the query “testicular germ cell tumors” AND “(protein-coding epigenetic players)”, with no time period restraints. Only papers that were written in English were considered. All abstracts were read in order to select those papers truly related to the topic.

Table 1 displays the result of our query, listing original articles addressing the role of these players in TGCTs pathogenesis and summarizing their major findings [34,35,36,37,38,39,40,41,42,43,44,45,46,47,48,49,50,51,52,53,54,55]. Despite the overwhelming evidence that stem cells and germ cells display dynamic epigenetic modifications during differentiation and spermatogenesis, including changes in the expression of these enzymes (e.g., with DNA methyltransferases more expressed in spermatogonia and histone methyltransferases mainly in spermatocytes) [10,56,57,58,59,60,61,62], there is still a lack of studies on the role of these players and related modifications in TGCTs (especially in certain families, with most studies published so far focusing on DNA-modifying enzymes).

In this line, we performed an in silico analysis of the publicly available The Cancer Genome Atlas (TCGA) database for TGCTs, regarding the diverse families of both DNA-modifying, histone modifying, and chromatin remodeling enzymes. We ultimately aimed to identify alterations in these players, exposing those potentially being the most relevant, and finally, providing the reader with a list of the most promising biomarkers to be further validated in independent patient cohorts. For this, we used the online resource cBioPortal for Cancer Genomics [63] and a user-defined entry gene set for all of these players. Statistical analysis with the available data was performed with Microsoft Excel 2016, GraphPad Prism 6 and IBM SPSS Statistics v.24. Distribution of continuous variables between groups was compared using the nonparametric Mann-Whitney *U* test. Co-occurrence/mutual exclusivity of alterations in pair of genes was assessed with the odds ratio (OR). Biomarker performance was assessed through the receiver operating characteristics (ROC) curve construction. ROC curves were constructed plotting sensitivity (true positive) against 1-specificity (false positive). A cut-off was established by the Youden’s index method [64,65]. Area under the curve (AUC) and biomarker performance parameters, including sensitivity, specificity, positive predictive value (PPV), negative predictive value (NPV), and accuracy, were ascertained. Survival curves were plotted with the Kaplan-Meier method and log-rank test was used for survival analysis. A *p*-value that was equal or inferior to 0.05 was considered to be significant. Bonferroni’s correction was applied to multiple pairwise comparisons.

A summary of major findings of this analysis is depicted in Table 2. 

### 3.1. DNA-MODIFYING ENZYMES

#### 3.1.1. Methylation

##### DNA Methyltransferases (*DNMTs*)

*DNMTs* are involved in many biological processes; they catalyze the transfer of a methyl group to DNA (both de novo or maintenance methylation), using S-adenosyl methionine (*SAM*) as the methyl-donor. There are three *DNMTs* with catalytic activity described in mammals: *DNMT1*, *DNMT3A,* and *DNMT3B* (*DNMT3L*, despite being structurally similar to *DNMT3A/3B*, is inactive on its own) [14]. These enzymes are deregulated in 27/156 (17%) TGCTs, mainly by mRNA upregulation (77.7% of the cases). The most commonly altered enzyme was *DNMT3B* (10% of tumor samples). *DNMT3A* and *DNMT3B* showed significantly co-occurrent alterations (logOR 2.785, adjusted *p*-value < 0.0001).

Regarding subtype discrimination, SEs disclosed significantly lower expression levels of all three enzymes (*p* < 0.0001) as compared to NSTs. The best performance was obtained for *DNMT3A*, rendering an AUC of 0.88. Interestingly, *DNMT3A* and *DNMT3B* expression was remarkably different among SEs and ECs (being strongly upregulated in the latter, *p* < 0.0001), with these enzymes discriminating among subtypes with AUCs of 0.98 and 0.99 (Figure 1A). Stage I patients also exhibited lower *DNMT3A* (*p* = 0.0006) and *DNMT3B* (*p* = 0.0011) expression levels when compared to stage II/III patients. No significant associations with overall (OS) or disease/progression-free (D/PFS) survival were depicted.

##### DNA Demethylases (*TETs*)

*TETs* catalyze the iterative demethylation of 5-methylcytosine (5mC). Three *TETs* are described in humans: *TET1-3* [23]. They are deregulated in 26/156 (17%) of TGCT samples, 80.8% of cases by mRNA upregulation. However, individually, alterations in these enzymes were present in less than 10% of the tumors, with the most commonly altered being *TET3* (in 8%).

Regarding subtype discrimination, SEs showed significantly higher expression levels of *TET2* when compared to NSTs (*p* < 0.0001), achieving an AUC = 0.79. Again, the differences in expression between SEs and ECs were quite remarkable (with upregulation in SEs, *p* < 0.0001), rendering an AUC = 0.98 (Figure 1B). Stage I disease also expressed significantly higher levels of *TET2* when compared to stages II/III (*p* = 0.0096). No significant associations with OS or D/PFS were depicted.

MAIN CONCLUSIONS: SEs display lower expression levels of *DNMTs* and higher expression levels of *TET2*, compatible with the described hypomethylated genome pattern of these tumors when compared to NSTs [29]. The expression pattern of these enzymes is completely opposite in ECs (with higher expression of *DNMTs* and lower expression of *TET2*), a finding that might prove useful in discriminating these two forms of TGCT, which have very different aggressiveness and prognosis. Also, there is room for prognostic impact of these markers, as *DNMTs*/*TET2* are upregulated and downregulated, respectively, in advanced stage disease. These findings are in accordance with most studies published so far, which also report *DNMTs’* overexpression in ECs [36,40,42,47,48,50] and of *TETs* in SEs [45] (Table 1).

### 3.2. HISTONE-MODIFYING ENZYMES

#### 3.2.1. Acetylation

##### Lysine Acetyltransferases (*KATs*)

###### A. MYST Family

The MYST family is the largest family of histone acetyl transferases (*HATs*), being responsible for acetylating the epsilon-amino group of lysine, direct PTM phenomena. *HATs* are, in general, qualified as transcription activators. The *MYST* family, specifically, is characterized by a distinct conserved domain, containing a C_2_HC zinc finger and an acetyl-CoA binding site. It includes five members in humans: *KAT5 (TIP60/PLIP)*, *KAT6A (MOZ/MYST3)*, *KAT6B (MORF/MYST4)*, *KAT7 (HBO1/MYST2)*, and *KAT8 (MOF/MYST1)* [66,67]. Globally, these genes are deregulated in 68/156 (44%) of TGCT samples. Most alterations consisted of mRNA upregulation (69%), followed by mRNA downregulation (22%). *KAT6A* was the member showing more frequent deregulation (in 20% of samples), followed by *KAT5* and *KAT7* (11% and 10% of cases, respectively). Alterations in *KAT6A* and *KAT7* were significantly mutually exclusive (*p* = 0.03, logOR < −3), but not after Bonferroni’s correction.

Regarding subtype discrimination, SEs showed significantly higher *KAT6A* and *KAT6B* expression levels (*p* < 0.0001 for both) when compared to NSTs. On the contrary, *KAT5* and *KAT8* were significantly downregulated in SEs as compared to NSTs (*p* < 0.0001 and *p* = 0.003, respectively). As a biomarker for discriminating SEs vs. NSTs, the best performance was rendered by *KAT5*, displaying an AUC = 0.75. Also, patients with stage I disease showed significantly higher expression levels of *KAT6B* and lower expression levels of *KAT8* when compared to stages II/III (*p* = 0.004 and *p* = 0.02).

###### B. GNAT Family

The *GNAT* (*GCN5*-related N-acetyltransferase) family is also involved in the reversible lysine acetylation of proteins such as histones and includes two main members, *KAT2A/GCN5* and *KAT2B/PCAF*, and also others like *ATAT1/MEC17*, *KAT1/HAT1*, *KAT9/ELP3*, and *AT1/SLC33A1.* They are characterized by sharing a domain with four conserved motifs A–D [24,68,69]. Globally, they are deregulated in 82/156 (53%) of TGCT samples, almost always due to mRNA upregulation (94%). The most commonly deregulated enzyme was *KAT9* (in 36% of TGCTs—43% of SEs and 28% of NSTs), the remainder only being deregulated in less than 10% of tumor samples. *KAT2A* and *KAT9* expression was found to be mutually exclusive (*p* = 0.026, logOR < −3), but it did not remain significant after Bonferroni’s correction.

*KAT1*, *KAT2A* (Figure 1C), *KAT2B,* and *KAT9* mRNA expression levels were significantly higher in SEs when compared to NSTs (*p* < 0.0001, *p* < 0.0001, *p* < 0.0001 and *p* = 0.0012), with the best discrimination performance disclosed by *KAT2A* (AUC = 0.78). On the other hand, *SLC33A1* and *ATAT1* were significantly downregulated in SEs as compared to NSTs (*p* < 0.0001, *p* = 0.0370). Also, *KAT2B* was significantly upregulated in patients with stage I disease compared to stages II/III (*p* = 0.0037). No impact on survival analysis was depicted.

###### C. Orphan Family

Besides the two aforementioned major *KATs* families, there are other enzymes with *HAT* activity, but which lack a true consensus *HAT* domain, and they are grouped together in the “orphan family” [70,71]. It includes the p300/CREB-binding protein pair (*KAT3A/CREBBP* and *KAT3B/EP300*), which have interchangeable roles during embryogenesis, and the nuclear and transcription factor-related *KATs* (such as *KAT4/TAF1*, *KAT12/GTF3C4*, *KAT13A/NCOA1*, *KAT13B/NCOA3*, *KAT13C/NCOA2*, and *KAT13D/CLOCK*). Globally, they are deregulated in 82/156 (53%) of TGCTs, mostly by mRNA upregulation (71%). The most commonly deregulated enzyme was *KAT13C* (in 41% of tumors, 94% of the cases by mRNA upregulation), followed by *KAT3B* (in 14% of tumors); also, both enzymes were displayed co-occurrent alterations, as did *KAT13A* and *KAT13C* (adjusted *p*-value < 0.001, logOR > 3, for both).

SEs showed significantly higher *KAT13A*, *KAT13C*, *KAT3B* expression levels (*p* < 0.0001 for all), and *CLOCK* (*p* = 0.0002). The best performance was depicted by *KAT13A* (AUC = 0.74), followed by *KAT13C* and *KAT3B* (AUC = 0.72). Also, stage I tumors showed the overexpression of *KAT3B, KAT13A*, and *KAT13C* as compared to stages II/III (*p* = 0.0128, *p* = 0.0308, *p* = 0.0304). No impact on survival analysis was depicted.

##### Lysine Deacetylases (*KDACs*)

###### A. Zn^2+^-Dependent Histone Deacetylases (HDACs)

*KDACs* target both histones and non-histone proteins, deacetylating their lysine residues, again contributing to PTM phenomena. They are generally regarded as transcriptional co-repressors. *KDACs* are organized into two major classes according to their dependence on co-factors: the Zn^2+^-dependent classical *HDACs* and the NAD^+^-dependent sirtuins’ family. Classical *HDACs* are usually grouped taking into account their basic structure, function, subcellular topography, and homology to yeast forms. To date, eleven *HDACs* have been identified in the human genome (*HDAC1–11*), which are assigned to four deacetylase classes: class I (*HDACs* 1, 2, 3 and 8), class IIa (*HDACs* 4, 5, 7 and 9), class IIb (*HDACs* 6 and 10), and class IV (*HDAC* 11) [72]. Globally, they are deregulated in 87/156 (56%) of TGCTs, 78% of the times due to mRNA upregulation. The most frequently altered proteins were *HDAC9* (14% of samples) and *HDAC1/HDAC7* (13% of samples); all *HDAC1* alterations consisted of mRNA upregulation. Significant co-occurrent alterations were shown between *HDAC6* and *HDAC8* (adjusted p-value 0.006, logOR 2.9).

Regarding subtype discrimination, *HDAC1, HDAC2, HDAC3, HDAC8, HDAC9,* and *HDAC11* (Figure 1D) were significantly downregulated in SEs when compared to NSTs (*p* = 0.0042 for *HDAC2*, *p* < 0.0001 for the remainder). On the contrary, *HDAC5*, *HDAC6*, and *HDAC7* were significantly upregulated in SEs (*p* = 0.0001, *p* < 0.0001, *p* = 0.0036). The discrimination performances were quite good, namely for *HDAC1* (AUC = 0.85), *HDAC8* (AUC = 0.85), *HDAC9* (AUC = 0.92), and *HDAC11* (AUC = 0.93). The latter allowed the following discriminating performance parameters: sensitivity = 82.4%, specificity = 92.3%, PPV = 93.3%, NPV = 80.0%, accuracy = 86.7%. Like for *DNMTs*/*TETs*, differences in expression of most *HDACs* were quite remarkable between SEs and ECs (with upregulation in the latter, *p* < 0.0001); for instance, *HDAC1* and *HDAC9* expression levels perfectly discriminated among these tumor subtypes with AUC=1. Also, stage II/III tumors showed significantly higher levels of *HDAC1* and *HDAC11* (*p* = 0.0002 and *p* = 0.0160) and lower levels of *HDAC9* (*p* = 0.0028). Nevertheless, there was no impact on survival.

###### B. NAD^+^-Dependent Sirtuin Deacetylases (SIRTs)

The remaining deacetylase class (class III) refers to the more recently uncovered *SIRT* family of proteins, which have the particularity of being dependent not on Zn^2+^ (thus being insensitive to hydroxamic acids that function as Zn^2+^-chelators), but on NAD^+^. There are seven *SIRTs* in the human genome (*SIRT1–7*) [73], which show deregulation in 53/156 (34%) of TGCTs, mainly by mRNA upregulation (87%). The most commonly deregulated enzymes were *SIRT2* and *SIRT6*, in 10% of samples. *SIRT3* and *SIRT6* were significantly concurrently altered (adjusted *p*-value 0.002, logOR > 3).

*SIRT4* and *SIRT5* expression was significantly lower and higher in SEs as compared to NSTs, respectively (*p* < 0.0001 for both); still, they rendered only modest AUCs of 0.77 and 0.72. Patients with stage II/III disease showed *SIRT4* overexpression (*p* = 0.01). No significant impact on survival was depicted.

MAIN CONCLUSIONS: SEs display higher expression levels of most acetylases and lower expression levels of most deacetylases, compatible with an acetylated, transcription-prone genome characteristic of these tumors. Again, important differences in the expression between SEs and NSTs (and especially between SEs and ECs) were noticed for *HDACs* (in accordance with the studies finding higher expression of *HDACs* in NST subtypes, such as choriocarcinoma and EC [53,54]), which might prove valuable in the clinical setting for discriminating these subtypes with different prognosis and treatment approaches. Regarding deacetylation, *HDACs* seem to have more impact in TGCTs biology than *SIRTs*. They were also found to associate with higher stage disease, as opposed to previous findings [53], meaning that studies in larger cohorts may be needed to ascertain their prognostic value.

#### 3.2.2. Methylation

##### Lysine Methyltransferases (*KMTs*)

###### A. SET Domain-Containing KMTs

Similar to methyltransferases that transfer methyl groups to DNA using *SAM* as a methyl donor, various enzymes catalyze this same transfer into histone proteins, specifically into lysine (and also arginine) residues. Depending on the residue and its position this change might provide transcriptional repression (H3K9 or H3K27, for example) or activation (H3K4, for instance). This major family of *KMTs* has a specific SET domain and it includes 51 members and various subfamilies, namely *SMYD*, *MLL*, *SET, EZH*, *SUV*, *PRDM*, and *NSD*-related proteins [74,75], and they are deregulated in 141/156 (90%) TGCT samples, mainly by mRNA upregulation (48%). However, 45% of tumors exhibited multiple alterations. The most commonly deregulated members were *SETD4* (45%), *EZH2* (21%), followed by *KMT2C/MLL3*, *NSD3,* and *PRDM10* (15% each). Five pairs of proteins showed significantly co-occurring alterations: *KMT2A* and *PRDM10* (logOR > 3, adjusted *p*-value 0.001), *NSD3* and *PRDM4* (logOR 2.4, adjusted *p*-value 0.012), *ASH1L* and *PRDM11* (logOR > 3, adjusted *p*-value 0.016), *ASH1L* and *SMYD1* (logOR > 3, adjusted *p*-value 0.016), and *KMT2D* and *PRDM4* (logOR 2.327, adjusted *p*-value 0.041).

Many proteins were differentially expressed among SEs and NSTs: SEs depicted significantly higher expression of *KMT2B*, *KMT2C*, *KMT2D*, *SETD1A*, *EZH1*, *SETDB2*, *SMYD3*, *PRDM2*, *PRDM15*, *PRDM1*, *PRDM7*, and *SETD4* (*p* < 0.0001), but lower expression of *EHMT2* (Figure 1E), *MECOM*, *SETD7*, *PRDM5*, (*p* < 0.0001), and *EZH2* (*p* = 0.0037), as compared to NSTs. The best discrimination power was rendered by *EHMT2/KMT1C*, *PRDM1*, *PRDM5* (all with AUC = 0.96), and *KMT2B* (AUC = 0.94). *EHMT2* displayed the following discrimination parameters: sensitivity = 94.1%, specificity = 96.9%, PPV = 97.6%, NPV = 92.6%, accuracy = 95.3%. Concerning associations with disease stage, the most impressive were *EHMT2*, which was significantly downregulated in stage I disease (*p* < 0.0001), and also *KMT2B* and *PRDM15*, which were significantly overexpressed in stage I disease (*p* = 0.0024 and *p* = 0.0002).

Regarding survival analysis, patients with altered *KMT2D/MLL2* showed significantly better D/PFS (*p* = 0.0284); also, patients with *PRDM2* alterations showed significantly worse OS and D/PFS (*p* = 0.0225 and *p* = 0.0432, respectively).

###### B. DOT1-Like Family (DOT1L)

*DOT1L* is the single member of this family of *KMTs*, which has a distinct structural domain [74,75,76]. It is deregulated in 14/156 (9%) TGCTs, mainly by mRNA upregulation (79%). Two missense mutations were found.

SEs exhibited significantly higher mRNA expression levels than NSTs (*p* < 0.0001), rendering an AUC = 0.79 for discriminating among both subtypes. Interestingly, *DOT1L* expression differed between SEs and pure ECs, with the former displaying higher levels (*p* < 0.0001); for these, an AUC = 0.87 was depicted. No association with the disease stage or survival was found.

##### Arginine Methyltransferases (*PRMTs*)

Another group of enzymes introduces methyl groups preferentially into arginine residues. There are nine *PRMTs* encoded in human genome (*PRMT1–9*) [77], and they show deregulation in 81/156 (52%) TGCTs, mainly by mRNA upregulation (50.6%) and also amplification (24.7%). The most commonly deregulated are *PRMT8* and *PRMT2*, in 21% and 13% of the samples, respectively; in particular, all alterations in *PRMT8* consisted of amplifications, except for one case with a missense mutation. No significant co-occurring or mutual exclusive alterations were found.

SEs showed significantly lower expression levels of *PRMT8* as compared to NSTs (*p* < 0.0001); *PRMT8* allowed for an AUC = 0.83 for discriminating both subtypes. On the contrary, *PRMT9* was overexpressed in SEs compared to NSTs, and an AUC = 0.75 was obtained. No associations with disease stage were depicted. However, patients with alterations in *PRMT4* (also known as *CARM1*) showed significantly poorer D/PFS (log rank, *p* = 0.003) (Figure 2A).

##### Lysine Demethylases (*KDMs*)

###### A. Alpha-Ketoglutarate (2OG) and Fe2^+^-Dependent, Jumonji (JmjC) Domain-Containing Demethylases

Regarding the removal of methyl groups from lysine residues, two classes of enzymes are considered, again based on their dependence of co-factors: the 2OG/Fe^2+^-dependent dioxygenases that contain a JmjC domain, and also the FAD-dependent amine oxidases. The former is the major family of *KDMs*, being composed of 29 different demethylase proteins [74,75,78], which are deregulated in 132/156 (85%) of TGCTs, mainly by mRNA upregulation (41%) and multiple alterations (40%). The most commonly altered enzyme was *KDM5A* (in 21% of TGCTs, by amplification in all but three tumors) and *KDM7A* (in 19% of TGCTs, always by mRNA upregulation). Two pairs, *KDM4D* + *KDM4E* and *JARID2* + *KDM2B*, tended to show co-occurring alterations (logOR > 3, adjusted *p*-value 0.014; and logOR 2.446, adjusted *p*-value 0.015).

SEs showed the overexpression of most demethylases, namely *KDM4D* (*p* = 0.0097), *KDM3A* (*p* = 0.0003), *KDM8, RIOX1* (Figure 1F), *HSPBAP1*, *KDM2A*, *KDM7A*, and *KDM3B* (*p* < 0.0001); however, the downregulation of *TYW5* and *JMJD8* (*p* < 0.0001 for both) was also disclosed. The best discrimination power was obtained with *HSPBAP1* (AUC = 0.93) and *RIOX1* (AUC = 0.89). Patients with stage I disease depicted higher *RIOX1* and *KDM2A* expression levels when compared to stages II/III (*p* = 0.0069 and *p* = 0.0441, respectively).

Regarding survival, patients with *KDM6B* and *PHF2* alterations endured significantly poorer OS (*p* = 0.00907) and D/PFS (*p* = 0.0150), respectively (Figure 2B).

###### B. FAD-Dependent Amine Oxidase Demethylases (LSDs)

The other class of KDMs contains two family members in this group: *KDM1A/LSD1* and *KDM1B/LSD2* [79]. They are deregulated in 22/156 (14%) TGCTs, mainly by mRNA upregulation (91%). The most commonly deregulated of the two was *LSD1* (8% of the samples). Alterations in these enzymes were neither mutually exclusive nor co-occurrent.

SEs overexpressed *LSD2* as compared to NSTs (*p* < 0.0001), but the discrimination power was rather modest (AUC = 0.73), and no differences were depicted for *LSD1*. Patients with stage I disease showed higher *LSD2* expression when compared to NSTs (*p* = 0.0062). No significant impact on survival was depicted.

MAIN CONCLUSIONS: SEs display higher expression levels of enzymes that establish activating modifications, such as *KDM4D*, *KDM3A*, *KMT2B/C/D*, *SETD1A*, and lower expression of those which establish repressive marks, like *EHMT2* and *EZH2* (despite the latter being reported in another work not to display significant differences in expression among SEs and NSTs [37]). More studies are needed to fully understand the interaction of all these enzymes, their respective modifications, and how they influence TGCTs biology.

#### 3.2.3. Phosphorylation

##### Serine/Threonine/Tyrosine Kinases

Besides methylation and acetylation, another histone PTMs can affect the chromatin structure. One of them is phosphorylation, which is introduced by proteins called kinases, and it has implications in many biological processes such as DNA repair and transcription regulation. These kinases, which include *RPS6KA3*, *RPS6KA4*, *RPS6KA5*, *ATR*, *ATM*, *BUB1*, *DCAF1*, *BAZ1B*, *MST1*, *HASPIN*, *JAK2*, *PRKDC*, *WEE1*, *AURKB*, *MAPK3*, *CHEK1*, *PKN1*, *CDK2*, *PAK2*, and *FYN*, [80] are deregulated in 134/156 (86%) of TGCTs, mainly by mRNA upregulation (63.4%). The only significantly co-occurring pair was *CHEK2* and *ATM* (logOR 2.565, adjusted *p*-value 0.029). The most frequently altered kinases were *BAZ1B* (34%) and *PRKDC* (33%).

Regarding differential expression among SEs vs NSTs, the most remarkable were *RPS6KA5*, *ATR*, *ATM*, and *BAZ1B* (overexpressed in SEs as compared to NSTs, *p* < 0.0001), and *RPS6KA4*, *PAK2*, and *AURKB* (downregulated in SEs compared to NSTs, *p* < 0.0001). The best discrimination was rendered by RPS6KA5 (AUC = 0.89) (Figure 1G), *BAZ1B* (AUC = 0.85), *AURKB*, and *RPS6KA4* (AUC = 0.83 for both). Patients with stage I disease displayed lower *AURKB* and *RPS6KA4* transcript levels; and higher *ATM* and *ATR* transcript levels when compared to stages II/III (*p* = 0.0373 and *p* = 0.0491; *p* = 0.0046 and *p* = 0.0078, respectively). Patients with *ATM* alterations showed poorer OS (*p* = 0.0468) and those with *RPS6KA4* and *PKN1* alterations disclosed poorer D/PFS (*p* = 0.0138 and *p* = 0.0361) (Figure 2C).

MAIN CONCLUSIONS: *ATM* and *AURKB*, the two kinases already studied in TGCTs [43,44], seem to have impact on TGCTs biology, showing differential expression between SEs and NSTs. More studies are needed to fully uncover the role of these enzymes in TGCTs.

#### 3.2.4. Ubiquitination

##### Ubiquitin Ligases

Histone ubiquitination (and deubiquitination) are less well explored PTMs, but they have been shown to crosstalk with the remaining modifications having impact on DNA repair and gene expression. The histone proteins most commonly conjugated with ubiquitin (especially monoubiquitination) are H2A and H2B. The enzymes catalyzing this reaction are ubiquitin ligases; they include *RING1*, *RNF2*, *BMI1*, *UBE2D3*, *RNF20*, *RNF40*, *UBE2A*, *UBE2B*, and *UBE2E1* [81,82] and are deregulated in 73/156 (47%) of TGCTs, mainly by mRNA upregulation (68%). The most frequently altered enzymes were *RING1* and *RNF40* (10% for both). No co-occurring or mutually exclusive pairs of enzymes with alterations were depicted.

SEs displayed significantly higher mRNA expression levels of *RNF2* and *BMI1* (Figure 1H) and lower expression of *RING1*, *RNF20*, and *UBE2A* when compared to NSTs (*p* < 0.0001 for all). The most remarkable enzyme in subtype discrimination was *BMI1*, reaching an AUC = 0.95, followed by *RNF20* (AUC = 0.90). Stage I patients also exhibited higher *BMI1* and *RNF2* expression (*p* = 0.0013, *p* = 0.352), and lower *RNF20* expression (*p* = 0.0127). Patients with *RNF40* and *UBE2E1* alterations disclosed poorer P/DFS (*p* = 0.0214 and *p* = 0.0282).

##### Deubiquitinating Enzymes

On the other hand, enzymes removing ubiquitin from histone residues are called deubiquitinating enzymes. The enzymes *USP16*, *USP21*, *MYSM1*, *BAP1*, *USP3*, and *USP22* [81] are deregulated in 58/156 (37%) TGCTs, mainly by mRNA upregulation (82.8%). The most frequently altered enzyme was *USP16* and *BAP1* (12% for both). Again, no co-occurring or mutually exclusive pairs of enzymes with alterations were depicted.

SEs showed significantly higher expression levels of *USP16*, reaching an AUC = 0.89; and, significantly lower levels of *BAP1* (Figure 1I), achieving an AUC = 0.84 (*p* < 0.0001). Also, stage I tumors displayed *USP16* overexpression (*p* = 0.0001) when compared to stages II/III.

MAIN CONCLUSIONS: Ubiquitination has not been explored thus far in TGCTs, but their differential expression among SEs and NSTs (reaching high AUC values) suggest that they might play an important role in tumorigenesis.

### 3.3. CHROMATIN REMODELING ENZYMES

ChRCs represent a wide range of proteins that have the common ability of inducing chromatin changes in a dynamic way, including nucleosome sliding, conformational modification of the nucleossome itself, and switching the composition of the histone octamers. Thus, they alter both histones and affect the histone-DNA interaction in the nucleosome. Through ATP hydrolysis, these players are grouped in four major families according to their core structure and presence of certain domains.

#### 3.3.1. *SWI/SNF* Family

The *SWI/SNF* chromatin remodelers contain bromodomains and they constitute a large complex composed of various subunits, including *BRG1/SMARCA4*, *BRM/SMARCA2*, *BAF180/PBRM1*, *ARID1A*, *ARID1B*, *ARID2*, *SNF5/SMARCB1*, *BRD7*, and *BAF60A/SMARCD1* [22,83]. They are deregulated in 84/156 (54%) TGCTs, mainly by mRNA upregulation (63%), but mRNA downregulation occurred in 13 cases. *ARID1B* showed significantly co-occurrent alterations with *PBRM1* (logOR > 3, adjusted *p*-value < 0.0001). The most frequently deregulated enzyme was *BAF60A/SMARCD1* (12%), *ARID2,* and *BRG1/SMARCA4* (11% for both).

SEs showed higher *BRG1* and *BRM* expression levels (*p* < 0.0001), but lower *SMARCD1* levels (*p* = 0.0020), as compared to NSTs. The best discrimination was achieved by *BRM*, rendering an AUC = 0.84. Stage I tumors showed significantly higher expression levels of *BRM* compared to stages II/III (*p* = 0.0004).

#### 3.3.2. *ISWI* Family

*SNF2H/SMARCA5*, *SNF2L/SMARCA1*, and *BAZ1A* are members of the *ISWI* family (which contain SANT-SLIDE modules) [22,84], and they show alterations in 32/56 (21%) TGCTs, mainly by mRNA upregulation (90.6%). *SNF2L* was the most frequently altered (15%), all consisting of mRNA upregulation. *SNF2L* and *BAZ1A* showed significantly co-occurrent alterations (logOR 2.201, adjusted *p*-value 0.005).

SEs exhibited significantly lower *SNF2L* expression levels when compared to NSTs (*p* < 0.0001) (Figure 1J), achieving an AUC = 0.96. Also, stage I patients showed lower *SNF2L* expression levels as compared to stages II/III (*p* = 0.0386). Patients with altered *SNF2L* showed significantly worse D/PFS (*p* = 0.0411) (Figure 2D).

#### 3.3.3. *CHD* Family

This family includes nine *CHD* enzymes (CHD1–9), which possess chromodomains [22,85]. They are deregulated in 92/156 (59%) TGCTs. Most alterations were due to mRNA upregulation (54.3%) and amplification (20.7%). The most frequently deregulated enzymes were *CHD7* (28%) and *CHD4* (21%). Three pairs tended to show co-occurrent alterations: *CHD3* and *CHD9* (logOR > 3, adjusted *p*-value < 0.001), *CHD2* and *CHD3* (logOR > 3, adjusted *p*-value 0.006), and *CHD2* and *CHD9* (logOR > 3, adjusted *p*-value 0.033).

Regarding subtype discrimination, SEs showed significantly higher expression levels of *CHD1*, *CHD2, CHD6,* and *CHD7* as compared to NSTs (*p* < 0.0001), while exhibiting lower *CHD4* expression (*p* < 0.0001). The best discrimination performance was achieved by *CHD1* and *CHD7* (AUC = 0.81 for both). Also, patients with stage I disease showed higher *CHD7* and *CHD8* transcript levels when compared to stages II/III (*p* = 0.0009 and *p* = 0.0026). Cases with *CHD8* and *CHD2* alterations showed poorer D/PFS (*p* = 0.0095 and *p* = 0.0493, respectively) (Figure 2E).

#### 3.3.4. *INO80* Family

*INO80*, *SWR1/SRCAP*, *RVB1/RUVBL1*, *RVB2/RUVBL2*, *YY1*, *ARP4/ACTL6A*, *ARP5/ACTR5*, and *ARP6/ACTR6* are members of the *INO80* family (which are characterized by helicase SANT domains) [22], and they are deregulated in 62/156 (40%) TGCTs, mainly by mRNA upregulation (74.2%). The most commonly deregulated enzyme was *ARP6/ACTR6* (12% of TGCTs). No single alteration in *ARP5/ACTR5* was depicted. There were no significant co-occurrent or mutually exclusive pairs.

Concerning differential expression among tumor subtype and disease stage, SEs and stage I patients displayed higher expression levels of *INO80* as compared to NSTs and stages II/III (*p* < 0.0001 and *p* = 0.0054, respectively), achieving an AUC = 0.88 for the SE vs. NST discrimination. Also, patients with *SRCAP* and *RUVBL2* alterations showed poorer D/PFS (*p* = 0.0488 and *p* = 0.0191) (Figure 2F).

MAIN CONCLUSIONS: Again, ChRCs represent unexplored territory in TGCTs. Alterations in *CHD* proteins are particularly frequent. This analysis points out they could be important not only in TGCT subtyping, but also in prognostication (survival impact).

## 4. Conclusions

The integrated molecular characterization of TGCTs is only now being uncovered [86]. There is an urgent need for better biomarkers that can supplant the classical serum markers used in clinical practice (which display many drawbacks), both for diagnostic, prognostic, and predictive purposes. DNA/histone-modifying enzymes (along with related modifications) and chromatin remodelers show promise as biomarkers, as they are frequently differentially expressed among the major classes and subtypes of TGCTs, reflecting the so-called developmental model of tumorigenesis and the locked epigenetic status of the corresponding cell of origin. Nevertheless, they are still scarcely explored in TGCT patients. In this work, we have analyzed the expression of several protein coding epigenetic enzymes at the mRNA level e tumor samples. Detection of such transcript-based biomarkers in liquid biopsies might be technically challenging; however, novel techniques for detection of circulating tumor cells and their transcripts are increasingly being employed with success in several tumor models and they should be pursued in TGCTs as well [87,88,89,90]. If effectively detected in liquid biopsies, these epigenetic players may be explored as biomarkers for targeted therapies. By allowing lower toxicity than the routinely employed chemotherapy regimens, these therapies might improve patients’ quality of life, which is fundamental for such young individuals with large life expectancy. Also, when used in combination, they may prove useful in overcoming cisplatin resistance, which eventually emerges in TGCT patients. The frequent upregulation of *DNMTs* in ECs (when compared to SEs) may, for instance, be used as a biomarker of susceptibility to *DNMT* inhibitors (DNMTi). These pharmacological agents comprehend both nucleoside and non-nucleoside analogs and the rationale for using them stands in the fact that by inhibiting the enzymatic activity of *DNMTs* they lead to the attenuation of malignant phenotype by inducing differentiation and tumor cell death (for review see [91]). Two of these agents (5-azacytidine and 5-aza-2′-deoxycytidine) are in fact already approved for treatment of patients with hematological malignancies, and they might prove useful in this particularly aggressive TGCT subtype. The frequent upregulation of *KDACs* in NSTs (the most aggressive and challenging subtypes of TGCTs), especially of those that are dependent on Zn^2+^ (*HDACs*), also indicates a potential benefit from *HDAC* inhibitors (HDACi), such as hydroxamic acid inhibitors (one of which—suberanilohydroxamic acid—is already approved again for the treatment of hematological cancers) [92,93]. Inhibitors of HATs (HATi), although not being particularly selective, may also aid in treating patients with SE which show frequently upregulation of these enzymes [94]. The use of such agents might allow for dose reduction of chemotherapy that these young patients endure, as SE is a highly chemo-sensitive solid tumor.

All in all, more studies in larger series are needed to explore the practical role and the clinical value of these enzymes in TGCTs.

## Figures and Tables

**Figure 1 cancers-11-00006-f001:**
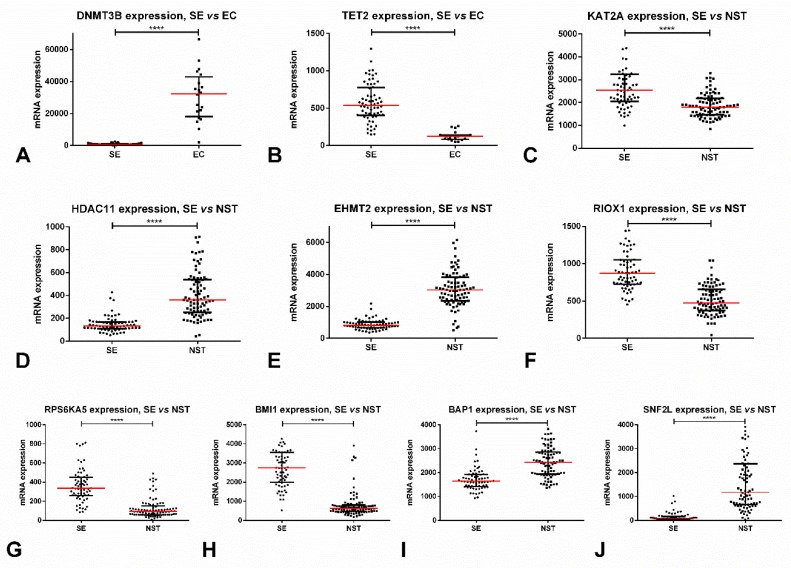
Most relevant alterations in protein-coding epigenetic players in testicular germ cell tumors, based on TCGA data. (**A**) Differential mRNA expression of DNMT3B among SE vs. EC; (**B**) Differential mRNA expression of TET2 among SE vs. EC; (**C**) Differential mRNA expression of KAT2A among SE vs. NST; (**D**) Differential mRNA expression of HDAC11 among SE vs. NST; (**E**) Differential mRNA expression of EHMT2 among SE vs. NST; (**F**) Differential mRNA expression of RIOX1 among SE vs. NST; (**G**) Differential mRNA expression of RPS6KA5 among SE vs. NST; (**H**) Differential mRNA expression of BMI1 among SE vs. NST; (**I**) Differential mRNA expression of BAP1 among SE vs. NST; (**J**) Differential mRNA expression of SNF2L among SE vs. NST. Abbreviations: EC—embryonal carcinoma; NST—non-seminomatous tumor; SE—seminoma; **** stands for *p* < 0.0001.

**Figure 2 cancers-11-00006-f002:**
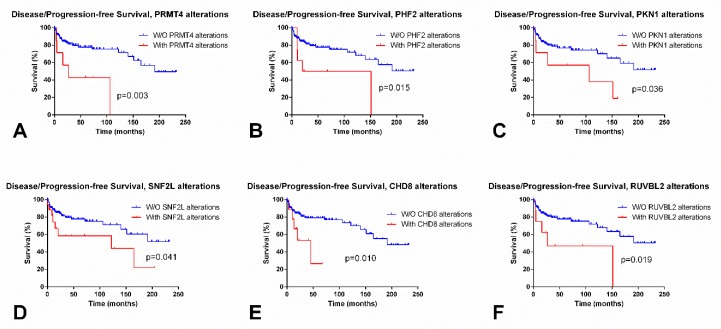
Most significant differences in disease/progression-free survival according to alterations in epigenetic enzymes in testicular germ cell tumors, based on TCGA data. (**A**) Disease/progression-free (D/PFS) according to alterations in PRMT4; (**B**) D/PFS according to alterations in PHF2; (**C**) D/PFS according to alterations in PKN1; (**D**) D/PFS according to alterations in SNF2L; (**E**) D/PFS according to alterations in CHD8; and, (**F**) D/PFS according to alterations in RUVBL2.

**Table 1 cancers-11-00006-t001:** Summary of most relevant publications regarding the role of DNA/histone-modifying enzymes and chromatin remodeling complexes involved in testicular germ cell tumors.

Family	Epigenetic Players	Major Findings	Sample Type and Size	Methodology	Author
Histone Kinases	*ATM*	pS-*ATM* constitutively detected in TGCTs (↑↑ in ECs) and normal testis	TGCTs and normal testes (FFPE)	IHC	Bartkova et al. 2005 [43]
	*AURKB*	↑ in SEs (vs. normal testis) correlated with Ki67 proliferation index	10 SEs (FFPE)	IHC	Chieffi et al. 2004 [44]
DNA Methyltransferases	*DNMT1*	*DNMT1* ↑↑ in ECs and ↑ in TEs (vs. absent in SEs);	32 TGCTs (FFPE)	RT-PCR IHC ISH	Omisanjo et al. 2007 [40]
		↑ in ECs (vs. normal testis) Evidence of a miR-199a/miR-214/*PSMD10/TP53/DNMT1* self-regulatory pathway	9 ECs + 12 normal testes (genomic RNA samples) TGCT cell lines	RT-PCR MSP WB	Chen et al. 2014 [48]
	*DNMT3A*	↑ in TGCTs (vs. normal testis) and associates with hypomethylation of intron 25	20 TGCTs + 9 adjacent tissue + 1 normal testis (FFPE and frozen tissue)	PCR IB/SB IHC	Ishii et al. 2007 [51]
		↑ in SEs (vs. normal testis)	8 SEs (frozen tissue)	Oligonucleotide-based microarray RT-PCR IHC	Yamada et al. 2004 [41]
		↑ in ECs (vs. normal testis); *DNMT3A* is the target of miR-199a-3p	11 TGCTs + 14 normal testes (genomic RNA samples) TGCT cell lines	RT-PCR MSP/BSP WB	Chen et al. 2014 [50]
	*DNMT3B*	↑ in stage III SEs; ↑ associates with poorer relapse-free survival	88 SEs (FFPE)	IHC	Arai et al. 2012 [39]
		↑ in ECs (vs. somatic solid tumors) ↑ leads to 5AZA hypersensitivity in ECs (vs. somatic tumors)	TGCT cell lines	RT-PCR WB	Beyrouthy et al. 2009 [36]
	*DNMT3L*	↑ in ECs (vs. other TGCT subtypes); ↑ in advanced stage SEs	TMAs (*n* = 83 TGCTs)	IHC	Matsuoka et al. 2016 [47]
		↑ in ECs (vs. somatic tumors and vs. other TGCT subtypes)	53 TGCTs (FFPE, 43 with frozen tissue) TGCT cell lines	RT-PCR IHC WB	Minami et al. 2010 [42]
Histone Methyltransferases	*EHMT2*	Its loss results in decreased tumor growth	TMA (67 TGCTs + 13 adjacent tissue + 4 normal testes) Cell lines Animal models	RT-PCR WB ChIP IHC	Ueda et al. 2014 [49]
	*EZH2*	↓ in GCNIS and TGCTs (vs. normal testis) No significant differences between SEs vs. NSTs	100 TGCTs + 4 GCNIS (frozen tissue)	RT-PCR	Hinz et al. 2010 [37]
		*EZH2* is expressed in GCNIS cells, but only in the cytoplasm	TGCTs and GCNIS (FFPE)	IHC	Almstrup et al. 2010 [34]
Histone Deacetylases	*HDAC1/2/3*	All 3 *HDACs* ↑ in CHs No associations with prognostic features	TMA (*n* = 325 TGCTs)	IHC	Fritzsche et al. 2011 [53]
	*HDAC1*	*HDAC1* is expressed at low levels in TGCTs	32 TGCTs (FFPE)	RT-PCR IHC ISH	Omisanjo et al. 2007 [40]
		*HDAC1* regulates EC cells proliferation by establishing H4K16; *HDAC1* ↑ in EC pluripotent cells (vs. non-pluripotent cells)	TGCT cell lines	RT-PCR ChIP WB	Yin et al. 2014 [54]
Histone Demethylases	*JMJD1A*	↓ in TGCTs (vs. normal testis); Its loss results in increased tumor growth	TMA (67 TGCTs + 13 adjacent tissue + 4 normal testes) Cell lines Animal models	RT-PCR WB ChIP IHC	Ueda et al. 2014 [49]
	*JMJD3*	*JMJD3* is absent in GCNIS	TGCTs and GCNIS (FFPE)	IHC	Almstrup et al. 2010 [34]
	*KDM6A*	*KDM6A* is absent in GCNIS	TGCTs and GCNIS (FFPE)	IHC	Almstrup et al. 2010 [34]
	*LSD1*	↑ in SEs (vs. normal testis) and pluripotent TGCT cells; *LSD1* inhibitors and *LSD1* knockdown impeded proliferation of pluripotent TGCT cells (vs. somatic tumors)	TMAs (*n* = 6 SEs) TGCT cell lines	WB IHC	Wang et al. 2011 [38]
		*LSD1* regulates EC cells proliferation by establishing H4K16; *LSD1* ↑ in EC pluripotent cells (vs. non-pluripotent cells)	TGCT cell lines	RT-PCR ChIP WB	Yin et al. 2014 [54]
Histone Methyltransferases	*PRDM1 and PRMT5*	H2AR3me2 and H4R3me2 establishment; ↑ co-expression in GCNIS and SEs (vs. NSTs) → silencing of differentiation-related genes	46 TGCTs + 15 GCNIS tissue (FFPE), 17 frozen tissue TGCT cell lines	Array RT-PCR Co-IP WB IHC	Eckert et al. 2008 [35]
	*PRDM2*	*PRDM2* binds ER-α and influences proliferation, survival and apoptosis	TGCT cell lines	RT-PCR WB IP	Zazzo et al. 2016 [55]
	*PRMT5*	*PRMT5* colocalizes with p44 (*AR* coactivator) *PRMT5* ↓ in the nucleus and ↑ in the cytoplasm (vs. normal testis)	33 SEs + 9 normal testes + 11 LCTs (FFPE)	IHC	Liang et al. 2006 [52]
DNA Demethylases	*TET1/2*	*TETs* ↓ in GCNIS	TGCTs + normal testes (FFPE and frozen tissue) TGCT cell lines	RT-PCR ELISA WB IHC/IF	Kristensen et al. 2014 [46]
	*TET1*	↑ in SEs (vs. normal testis and vs. NSTs)	47 TGCTs + 7 normal testes (frozen tissue) TGCT cell lines	RT-PCR Droplet digital PCR IHC	Benesova et al. 2017 [45]

Upward (↑) and downward (↓) arrows stand for up- and downregulation, respectively. Abbreviations: AR—androgen receptor; BSP—bisulfite sequencing PCR; CH—choriocarcinoma; ChIP—chromatin immunoprecipitation; CoIP—co-immunoprecipitation; EC—embryonal carcinoma; ER-α—estrogen receptor alpha; FFPE—formalin-fixed paraffin embedded; GCNIS—germ cell neoplasia in situ; IB—immunoblot; IF—immunofluorescence; IHC—immunohistochemistry; ISH—in situ hybridization; LCT—Leydig cell tumor; MSP—methylation specific PCR; NST—non-seminomatous tumor; PCR—polymerase chain reaction; pS-ATM—S1981-phosphorylated ATM; RT-PCR—real-time PCR; SB—southern blot; SE—seminoma; TGCT—testicular germ cell tumor; TMA—tissue microarray; WB—western blot; 5AZA—5-aza-2′deoxycytidine.

**Table 2 cancers-11-00006-t002:** Summary of most relevant DNA/histone-modifying enzymes and chromatin remodeling complexes involved in testicular germ cell tumors according to The Cancer Genome Atlas (TCGA) database analysis.

PLAYERS *	Most Frequently Deregulated (% of Cases)	Related Alterations (logOR)	SEvsNST	Best Performance (AUC)	Association with Stage	Survival Impact
**DNA-modifying enzymes**						
*DNMTs*	*DNMT3B* (10)	*DNMT3A* and *DNMT3B* (co-occurrent, 2.785)	All ↓ in SE	*DNMT3A* (0.88)	Yes (*DNMT3A/3B*)	No
*TETs*	*TET3* (8)	No	*TET2* ↑ in SE	*TET2* (0.79)	Yes (*TET2*)	No
**Histone-modifying enzymes**						
ACETYLATION						
KATs						
*MYST* family	*KAT6A* (20)	*KAT6A* and *KAT7* (mutually exclusive, <−3)	*KAT6A/6B* ↑ in SE; *KAT5/8* ↓ in SE	*KAT5* (0.75)	Yes (*KAT6B* and *KAT8*)	No
*GNAT* family	*KAT9* (36)	*KAT2A* and *KAT9* (mutually exclusive, <−3)	*KAT1/2A/2B/9* ↑ in SE; *SLC33A1* and *ATAT1* ↓ in SE	*KAT2A* (0.78)	Yes (*KAT2B*)	No
*Orphan* family	*KAT13C* (41)	*KAT3B* and *KAT13C*; *KAT13A* and *KAT13C* (co-occurrent, >3)	*KAT3B/13A/13C* and *CLOCK* ↑ in SE	*KAT13A* (0.74)	Yes (*KAT3B/13A/13C*)	No
KDACs						
*HDACs*	*HDAC9* (14)	*HDAC6* and *HDAC8* (co-occurrent, 2.9)	*HDAC1/2/3/8/9/11* ↓ in SE; *HDAC5/6/7* ↑ in SE	*HDAC11* (0.93)	Yes (*HDAC1/11*)	No
*SIRTs*	*SIRT2/6* (10)	*SIRT3* and *SIRT6* (co-occurrent, >3)	*SIRT4* ↓ in SE; *SIRT5* ↑ in SE	*SIRT4* (0.77)	Yes (*SIRT4*)	No
METHYLATION						
KMTs						
*SET* domain	*SETD4* (45) *EZH2* (21)	*KMT2A* and *PRDM10*; *ASH1L* and *PRDM11*; *ASH1L* and *SMYD1* (co-occurrent, >3)	*KMT2B/2C/2D*, *SETD1A/D4/DB2*, *EZH1, SMYD3* and *PRDM1/2/7/15* ↑ in SE; *EHMT2, MECOM, SETD7, PRDM5* and *EZH2* ↓ in SE	*EHMT2, PRDM1* and *PRDM5* (AUC 0.96)	Yes (*EHMT2, KMT2B* and *PRDM15*)	Yes (*KMT2D* and *PRDM2*)
*DOT1-like*	*DOT1L* (9)	N/A	↑ in SE	*DOT1L* (0.79)	No	No
PRMTs	*PRMT8* (21)	No	*PRMT8* ↓ in SE *PRMT9* ↑ in SE	*PRMT8* (0.83)	No	Yes (*PRMT4*)
KDMs						
*Jumonji*-domain	*KDM5A* (21) *KDM7A* (19)	*KDM4D* and *KDM4E* (co-occurrent, >3)	*KDM2A/3A/3B/4D/7A/8*, *RIOX1* and *HSPBAP1* ↑ in SE; *TYW5* and *JMJD8* ↓ in SE	*HSPBAP1* (0.93)	Yes (*RIOX1* and *KDM2A*)	Yes (*KDM6B* and *PHF2*)
*LSDs*	*LSD1* (8)	No	*LSD2* ↑ in SE	*LSD2* (0.73)	Yes (*LSD2*)	No
PHOSPHORYLATION						
Kinases	*BAZ1B* (34) *PRKDC* (33)	*CHEK2* and *ATM* (co-occurrent, 2.565)	*BAZ1B, ATM, ATR* and *RPS6KA5* ↑ in SE; *RPS6KA4, PAK2* and *AURKB* ↓ in SE	*RPS6KA5* (0.89)	Yes (*AURKB*, *ATM*, *ATR* and *RPS6KA4*)	Yes (*ATM*, *RPS6KA4* and *PKN1*)
UBIQUITINATION						
Ubiquitin ligases	*RING1* and *RNF40* (10)	No	*RNF2* and *BMI1* ↑ in SE; *RING1, RNF20* and *EBE2A* ↓ in SE	*BMI1* (0.95)	Yes (*BMI1*, *RNF2* and *RNF20*)	Yes (*RNF40* and *UBE2E1*)
Deubiquitinating enzymes	*USP16* and *BAP1* (12)	No	*USP16* ↑ in SE; *BAP1* ↓ in SE	*USP16* (0.89)	Yes (*USP16*)	No
**Chromatin remodeling complexes**						
SWI/SNF	*SMARCD1* (12)	*ARID1B* and *PBRM1* (co-occurrent, >3)	*BRG1* and *BRM* ↑ in SE; *SMARCD1* ↓ in SE	*BRM* (0.84)	Yes (*BRM*)	No
ISWI	*SNF2L* (15)	*SNF2L* and *BAZ1A* (co-occurrent, 2.201)	*SNF2L* ↓ in SE	*SNF2L* (0.96)	Yes (*SNF2L*)	Yes (*SNF2L*)
CHD	*CHD7* (28) *CHD4* (21)	*CHD3*, *CHD2* and *CHD9* (co-occurrent alterations, >3)	*CHD1/2/6/7* ↑ in SE; CHD4 ↓ in SE	*CDH1* and *CHD7* (0.81)	Yes (*CHD7* and *CHD8*)	Yes (*CHD2* and *CHD8*)
INO80	*ARP6* (12)	No	*INO80* ↑ in SE	*INO80* (0.88)	Yes (*INO80*)	Yes (*SRCAP* and *RUVBL2*)

* Only the most relevant and significant players are represented. Upward (↑) and downward (↓) arrows stand for up- and downregulation, respectively. Abbreviations: AUC—area under the curve; N/A—not applicable; NST—non-seminomatous tumors; OR—odds ratio; SE—seminoma; TGCT—testicular germ cell tumors.

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
