# Peer review of "The Role of DNA/Histone Modifying Enzymes and Chromatin Remodeling Complexes in Testicular Germ Cell Tumors"

_cancers, 2018, doi:10.3390/cancers11010006_

Reviewer 1 Report

The authors analyzed the proteins involved in chromatin remodelling in testicular germ cell tumors.

It is necessary an extensive revisiono f english form a nd also of the biology’s nomenclature used.

The structure of the review should be completely modified.  For example, some of the information reported in the results section could be reported  into the table. Authors should focue thier attention not only on the evidence but also on the implications of their results.

A discussion section is needed to correlate the results obtained analyzing “The Cancer Genome Atlas” with data previously obtained in in TGCT and in other tumors.

I think that the queries used to select the papers should be revised because a lot of recently published manuscripts are excluded.

 I reported a list of manuscpripts that should be added:

An up-date on epigenetic and molecular markers in testicular germ cell tumors.

Chieffi P.

Intractable Rare Dis Res. 2017 Nov;6(4):319-321. doi: 10.5582/irdr.2017.01070.

Epigenetic and risk factors of testicular germ cell tumors: a brief review.

Landero-Huerta DA, Vigueras-Villasenor RM, Yokoyama-Rebollar E, Arechaga-Ocampo E, Rojas-Castaneda JC, Jimenez-Trejo F, Chavez-Saldana M.

Front Biosci (Landmark Ed). 2017 Mar 1;22:1073-1098. Review.

Critical Function of PRDM2 in the Neoplastic Growth of Testicular Germ Cell Tumors.

Di Zazzo E, Porcile C, Bartollino S, Moncharmont B.

Biology (Basel). 2016 Dec 14;5(4). pii: E54.

PR/SET Domain Family and Cancer: Novel Insights from the Cancer Genome Atlas.

Sorrentino A, Federico A, Rienzo M, Gazzerro P, Bifulco M, Ciccodicola A, Casamassimi A, Abbondanza C.

PRDM Proteins: Molecular Mechanisms in Signal Transduction and Transcriptional Regulation.

Di Zazzo E, De Rosa C, Abbondanza C, Moncharmont B.

Biology (Basel). 2013 Jan 14;2(1):107-41. doi: 10.3390/biology2010107.

Author Response

The authors analyzed the proteins involved in chromatin remodeling in testicular germ cell tumors. It is necessary an extensive revision of English form and also of the biology’s nomenclature used.

Reply: We thank the reviewer for this comment. Accordingly, we have reviewed the English and changed some phrases/expressions. Changes are highlighted in red throughout the text. Gene names have been italicized, as also pointed out by Reviewer #2.

The structure of the review should be completely modified.  For example, some of the information reported in the results section could be reported into the table. Authors should focus their attention not only on the evidence but also on the implications of their results.

Reply: We thank the Reviewer for his/her comment and for allowing us to clarify.

Regarding the translation of information of the analyses into a Table, the authors have already performed this (Table 2). This Table provides a good summary of the major alterations reported in every family of enzymes.

Regarding the focus on the implications of the results: the authors would like to state that the aim of the Review was to analyze the publicly available data contained in TCGA database for TGCTs (which has not yet been picked up and explored) and to thoroughly describe alterations in all families of epigenetic enzymes and chromatin remodelers, this way exposing the ones potentially being the most relevant. The ultimate aim of the paper is to provide the reader with a list of potential enzymes, which can then be used by research groups for validation in their own cohorts of patient samples, eventually confirming their clinical value. As an example, our own research group has already validated some findings concerning two of these major families of epigenetic enzymes (unpublished data) in our own cohort and in vitro. This way, we believe that the main organization/structure of the Review suits this purpose. Also, such organization was valued by the other Reviewers (see below), and “completely modifying” it as suggested would enter in conflict with the other comments.

We have, nevertheless, better clarified the major aim/intent of the Review: “We ultimately aimed to identify alterations in these players, exposing those potentially being the most relevant and, finally, providing the reader with a list of the most promising biomarkers to be further validated in independent patient cohorts.

A discussion section is needed to correlate the results obtained analyzing “The Cancer Genome Atlas” with data previously obtained in TGCT and in other tumors.

Reply: We thank the Reviewer for his/her comment and for allowing us to clarify this issue. Regarding the “discussion section” – such section was, indeed, already included by us. In fact, at the end of each section concerning a major family of epigenetic enzymes, we have included one or two paragraphs briefly discussing the most relevant above-mentioned findings resulting from the analysis of the TCGA database, and also commented on studies (included in Table 1) that validate or somewhat are in consonance with these findings. This way, the authors have avoided a formal “discussion section chapter” at the end of the analysis, which would mean that the reader would have to come back to the analysis in order to read the interpretation of results. This way, we believe it became a lot more “reader-friendly”, as the reader can go throughout a family of enzymes, read the analyses and then immediately see the comparison with published literature and have some biological insight about the findings.

Regarding the comparison with other studies on TGCTs: as said above, the authors have already performed that, both in Table 1 and in the mini-discussion paragraphs provided at the end of each section. Still, the authors have added some more detailed information:

These findings are in accordance with most studies published so far, which also report DNMTs’ overexpression in ECs [36, 40, 42, 47, 48, 50] and of TETs in SEs [45] (Table 1).

Again, important differences in expression between SEs and NSTs (and especially between SEs and ECs) were noticed for HDACs (in accordance with the studies finding higher expression of HDACs in NST subtypes such as choriocarcinoma and EC [53-54]), which might prove valuable in the clinical setting for discriminating these subtypes with different prognosis and treatment approaches. Regarding deacetylation, HDACs seem to have more impact in TGCTs biology than SIRTs. They were also found to associate with higher stage disease, as opposed to previous findings [53], meaning studies in larger cohorts may be needed to ascertain their prognostic value.

…EZH2 (despite the latter being reported in another work not to display significant differences in expression among SEs and NSTs [37]).

Regarding the comparison with findings from other tumors: as the Reviewer surely acknowledges, TGCTs are truly at the crossroads between cancer and developmental biology, reflecting the epigenetic status of the cell of origin in a developmental manner. They have, then, a locked epigenetic status and specificities in their chromatin pattern, which are very distinct from other somatic cancers. The comparison of expression or deregulation of epigenetic enzymes in such biologically distinct tumor models would, in our opinion, lead to losing focus on TGCTs and the reader would perhaps not retrieve much information from this kind of analyses. Also, the paper is already quite extensive in its original form.

I think that the queries used to select the papers should be revised because a lot of recently published manuscripts are excluded. I reported a list of manuscripts that should be added: 

An up-date on epigenetic and molecular markers in testicular germ cell tumors.

Chieffi P. Intractable Rare Dis Res. 2017 Nov;6(4):319-321. doi: 10.5582/irdr.2017.01070.

Epigenetic and risk factors of testicular germ cell tumors: a brief review. Landero-Huerta DA, Vigueras-Villasenor RM, Yokoyama-Rebollar E, Arechaga-Ocampo E, Rojas-Castaneda JC, Jimenez-Trejo F, Chavez-Saldana M. Front Biosci (Landmark Ed). 2017 Mar 1;22:1073-1098. Review.

Critical Function of PRDM2 in the Neoplastic Growth of Testicular Germ Cell Tumors.

Di Zazzo E, Porcile C, Bartollino S, Moncharmont B. Biology (Basel). 2016 Dec 14;5(4). pii: E54.

PR/SET Domain Family and Cancer: Novel Insights from the Cancer Genome Atlas.

Sorrentino A, Federico A, Rienzo M, Gazzerro P, Bifulco M, Ciccodicola A, Casamassimi A, Abbondanza C.

PRDM Proteins: Molecular Mechanisms in Signal Transduction and Transcriptional Regulation. Di Zazzo E, De Rosa C, Abbondanza C, Moncharmont B. Biology (Basel). 2013 Jan 14;2(1):107-41. doi: 10.3390/biology2010107.

Reply: We thank the Reviewer for suggesting these papers, of which we naturally are aware of and have carefully read before writing our manuscript. However, considering the aim of our Review (and hence its query entrance for searching papers on Medline) we have almost exclusively focused on Original Papers that have explored the role of epigenetic enzymes in TGCTs. The Reviewer, however, has indicated a list of 5 papers to be cited, 3 of which correspond to broad Review papers and 1 corresponding to a in silico analysis of the TCGA database in a pan-cancer perspective; so, we believe that our query entrance is correct, since it only addressed Original Papers, suiting the aim of the Review. Nevertheless, we thank the Reviewer for pointing out the paper “Critical Function of PRDM2 in the Neoplastic Growth of Testicular Germ Cell Tumors”, which indeed was missed during our search. We have included it on the manuscript, and on Table 1, so that the reader might compare with the in silico analysis.

Furthermore, we have clarified the query entrance: “Table 1 displays the result of our query, listing studies (original articles)”.

Reviewer 2 Report

In their manuscript the authors review on the molecular mechanisms underlying Testicular Germ Cell Tumors (TGCTs) and analyze publicly available data to evaluate the potential of DNA/histone modifying enzymes as predictive markers in this disease. The authors have used a comprehensive approach entailing both investigation of literature and in silico analysis. Their findings on this topic is relevant to the field and will be of high interest to the readership of “Cancers”. A number of minor points should be addressed prior to publication as detailed below.

1.       In Table 1, a summary of the most relevant publications on enzymatic factors involved in TGCTs is presented. The findings are ordered according to the date of publication (ascending date in the last column). Since some enzymes are mentioned more than once (e.g. DNMT3L), the reviewer would suggest to order the rows alphabetically based on the enzyme. This will enable the reader to compare the outcomes of studies on the same enzyme. For instance, the readers will be able to compare whether both studies on DNMT3L concluded that this enzyme is upregulated in embryonal carcinoma. In addition, one extra column containing information on the identity/function of the enzyme would be helpful, e.g. methyltransferase, acetyltransferases, etc.

2.       It is not clear on which basis the authors decided which data will be shown in Figure 1. In the beginning it is comprehensible why Figure 1A and 1B are shown since they both compare the expression of a DNA methyltransferase and DNA methylase, respectively, in seminomas and embryonal carcinomas. However, the graphs of Figure 1 do not represent all the enzyme classes mentioned in this manuscript. For instance, there are two figures for HDAC11 (Figure 1C-D) but there are no graphs for ubiquitin ligases/deubiquitinating enzymes. Together, it seems like it was randomly decided which data will be shown in a graph. To provide a general overview, the authors should provide at least one representative and significant image demonstrating the most important finding of each enzyme category (i.e. DNA methyltransferases, DNA demethylases, histone methyltransferases, histone demethylases, histone acetyltransferases, histone deacetylases, kinases, (de-)ubiquitinating enzymes, chromatin remodelers).

3.       It would be helpful if the authours would add a short description of the general function of each presented enzyme class (1-2 sentences).

4.       The style is not always consistent (e.g. testicular germ cell tumors [line 22] and Testicular Germ Cell Tumors [line 151]).

5.       Human gene names should be written in capital letters and be italicized. Moreover, “in silico” (line 99) should be italicized.

6.       The manuscript has some typing and grammatical errors which need to be corrected (e.g. Bonferroni’s correctio”; line 113).

Author Response

Reviewer #2:

In their manuscript the authors review on the molecular mechanisms underlying Testicular Germ Cell Tumors (TGCTs) and analyze publicly available data to evaluate the potential of DNA/histone modifying enzymes as predictive markers in this disease. The authors have used a comprehensive approach entailing both investigation of literature and in silico analysis. Their findings on this topic is relevant to the field and will be of high interest to the readership of “Cancers”.

Reply: We thank the reviewer for his/her positive opinion on our manuscript. Additionally, we sincerely want to thank the Reviewer for his/her comments, with which we agree, and which allowed us to improved Manuscript.

A number of minor points should be addressed prior to publication as detailed below.

In Table 1, a summary of the most relevant publications on enzymatic factors involved in TGCTs is presented. The findings are ordered according to the date of publication (ascending date in the last column). Since some enzymes are mentioned more than once (e.g. DNMT3L), the reviewer would suggest to order the rows alphabetically based on the enzyme. This will enable the reader to compare the outcomes of studies on the same enzyme. For instance, the readers will be able to compare whether both studies on DNMT3L concluded that this enzyme is upregulated in embryonal carcinoma. In addition, one extra column containing information on the identity/function of the enzyme would be helpful, e.g. methyltransferase, acetyltransferases, etc.

Reply: We would like to thank the Reviewer for his/her advice, with which we do agree. We have changed Table 1 accordingly and added the suggested extra column.

It is not clear on which basis the authors decided which data will be shown in Figure 1. In the beginning it is comprehensible why Figure 1A and 1B are shown since they both compare the expression of a DNA methyltransferase and DNA methylase, respectively, in seminomas and embryonal carcinomas. However, the graphs of Figure 1 do not represent all the enzyme classes mentioned in this manuscript. For instance, there are two figures for HDAC11 (Figure 1C-D) but there are no graphs for ubiquitin ligases/deubiquitinating enzymes. Together, it seems like it was randomly decided which data will be shown in a graph. To provide a general overview, the authors should provide at least one representative and significant image demonstrating the most important finding of each enzyme category (i.e. DNA methyltransferases, DNA demethylases, histone methyltransferases, histone demethylases, histone acetyltransferases, histone deacetylases, kinases, (de-)ubiquitinating enzymes, chromatin remodelers).

Reply: We thank the Reviewer for his/her valuable suggestion. Accordingly, we have constructed a new Figure 1, corresponding to a panel of 10 graphs, one per each major family of enzymes, illustrating some of the most remarkable findings resulting from the analyses.                                        

It would be helpful if the authors would add a short description of the general function of each presented enzyme class (1-2 sentences).

Reply: We thank the Reviewer for his/her suggestion. Accordingly, we have added a short sentence in each section describing the general function of each enzyme class.

“DNMTs are involved in many biological processes; they catalyze the transfer of a methyl group to DNA (both de novo or maintenance methylation), using S-adenosyl methionine (SAM) as the methyl-donor.

TETs catalyze the iterative demethylation of 5-methylcytosine (5mC).

The MYST family is the largest family of histone acetyl transferases (HATs), being responsible for acetylating the epsilon-amino group of lysine, direct PTM phenomena. HATs are, in general, qualified as transcription activators. The MYST family, specifically, is characterized by a distinct conserved domain, containing a C2HC zinc finger and an acetyl-CoA binding site.

The GNAT (GCN5-related N-acetyltransferase) family is also involved in the reversible lysine acetylation of proteins such as histones and includes two main members, KAT2A/GCN5 and KAT2B/PCAF, and others like ATAT1/MEC17, KAT1/HAT1, KAT9/ELP3 and AT1/SLC33A1. They are characterized by sharing a domain with four conserved motifs A-D. KDACs target both histones and non-histone proteins, deacetylating their lysine residues, again contributing to PTM phenomena. They are generally regarded as transcriptional co-repressors. KDACs are organized into two major classes according to their dependence on co-factors: the Zn2+-dependent classical HDACs and the NAD+-dependent sirtuin family (see below). Classical HDACs are usually grouped taking into account their basic structure, function, subcellular topography and homology to yeast forms. To date, eleven HDACs have been identified in the human genome (HDAC1-11), which are assigned to four deacetylase classes: class I (HDACs 1, 2, 3 and 8), class IIa (HDACs 4, 5, 7 and 9), class IIb (HDACs 6 and 10) and class IV (HDAC 11)

The remaining deacetylase class (class III) refers to the more recently uncovered SIRT family of proteins, which have the particularity of being dependent not on Zn2+ (thus being insensitive to hydroxamic acids which function as Zn2+-chelators), but on NAD+.

Similar to methyltransferases that transfer methyl groups to DNA using SAM as a methyl donor, various enzymes catalyze this same transfer into histone proteins, specifically into lysine (and also arginine, see below) residues. Depending on the residue and its position this change might provide transcriptional repression (H3K9 or H3K27, for example) or activation (H3K4, for instance).

Another group of enzymes introduces methyl groups preferentially into arginine residues.

Regarding the removal of methyl groups from lysine residues, two classes of enzymes are considered, again based on their dependence of co-factors: the 2OG/Fe2+-dependent dioxygenases which contain a JmjC domain, and also the FAD-dependent amine oxidases (see below). The former is the major family of KDMs

Besides methylation and acetylation, other histone PTMs can affect chromatin structure. One of them is phosphorylation, which is introduced by proteins called kinases, and has implications in many biological processes such as DNA repair and transcription regulation.

Histone ubiquitination (and deubiquitination) are less well explored PTMs, but have been shown to crosstalk with the remaining modifications having impact on DNA repair and gene expression. The histone proteins most commonly conjugated with ubiquitin (especially monoubiquitination) are H2A and H2B. The enzymes catalyzing this reaction are ubiquitin ligases;

On the other hand, enzymes removing ubiquitin from histone residues are called deubiquitinating enzymes.

Chromatin remodelers represent a wide range of proteins which have the common ability of inducing chromatin changes in a dynamic way, including nucleosome sliding, conformational modification of the nucleosome itself and switching the composition of the histone octamers. Thus, they alter both histones and affect the histone-DNA interaction in the nucleosome. Through ATP hydrolysis. These players are grouped in four major families according to their core structure and presence of certain domains.”

The style is not always consistent (e.g. testicular germ cell tumors [line 22] and Testicular Germ Cell Tumors [line 151]).

Reply: We thank the Reviewer for calling our attention to this. We have corrected it accordingly.

Human gene names should be written in capital letters and be italicized. Moreover, “in silico” (line 99) should be italicized.

Reply: We thank the Reviewer for calling our attention to this. We have italicized gene names throughout the manuscript and corrected “in silico”.

The manuscript has some typing and grammatical errors which need to be corrected (e.g. Bonferroni’s correctio”; line 113).

Reply: We thank the Reviewer for pointing this out. We have corrected these typos accordingly.

Reviewer 3 Report

The manuscript is a nice summary of DNA modifying enzymes in testicular germ cell tumors. A few suggestions to improve the manuscript.

1) Survival curves for the most important gene in each category would be helpful.

2) As mentioned in the conclusions, some of these genes may be potential biomarkers of therapy targets. Point out in a table or in the text, which genes have the potential to be found in biofluids and can be used as biomarkers.

3) It would also be helpful to analyze data from other sources such as the GDC data portal.

Author Response

The manuscript is a nice summary of DNA modifying enzymes in testicular germ cell tumors.

Reply: We thank the reviewer for his/her positive opinion on our manuscript.

A few suggestions to improve the manuscript.

Survival curves for the most important gene in each category would be helpful.

Reply: We thank the Reviewer for his suggestion. As Reviewer #2 also suggested including one graph per each family of enzymes (which we did in Figure 1), adding another similar panel containing survival curves for all the 10 families of enzymes would perhaps be very extensive, especially because in some classes no significant impact on survival was depicted. In this line, and to follow the Reviewer’s suggestion, we have created a new Figure 2, containing only survival curves for some of the most representative enzymes (6 enzymes), in which the most significant impact on survival was found.      

As mentioned in the conclusions, some of these genes may be potential biomarkers of therapy targets. Point out in a table or in the text, which genes have the potential to be found in biofluids and can be used as biomarkers.

Reply: We thank the Reviewer for his/her comment. In accordance, we have added a section about this matter in the Conclusion: “DNA/histone modifying enzymes (along with related modifications) and chromatin remodelers show promise as biomarkers as they are frequently differentially expressed among the major classes and subtypes of TGCTs, reflecting the so-called developmental model of tumorigenesis and the locked epigenetic status of the corresponding cell of origin. Nevertheless, they are still scarcely explored in TGCT patients. In this work we have analyzed the expression of several protein coding epigenetic enzymes at the mRNA level e tumor samples. Detection of such transcript-based biomarkers in liquid biopsies might be technically challenging; however, novel techniques for detection of circulating tumor cells and their transcripts are increasingly being employed with success in several tumor models, and should be pursued in TGCTs as well [87-90]. If effectively detected in liquid biopsies these epigenetic players may be explored as biomarkers for targeted therapies. By allowing lower toxicity than the routinely employed chemotherapy regimens these therapies might improve patients’ quality of life, which is fundamental for such young individuals with large life expectancy. Also, when used in combination, they may prove useful in overcoming cisplatin resistance, which eventually emerges in TGCT patients. The frequent upregulation of DNMTs in ECs (when compared to SEs) may, for instance, be used as a biomarker of susceptibility to DNMT inhibitors (DNMTi). These pharmacological agents comprehend both nucleoside and non-nucleoside analogs and the rationale for using them stands on the fact that by inhibiting the enzymatic activity of DNMTs they lead to attenuation of malignant phenotype by inducing differentiation and tumor cell death (for review see [91]). Two of these agents (5-azacytidine and 5-aza-2’-deoxycytidine) are in fact already approved for treatment of patients with hematological malignancies, and might prove useful in this particularly aggressive TGCT subtype. The frequent upregulation of KDACs in NSTs (the most aggressive and challenging subtypes of TGCTs), especially of those dependent on Zn2+ (HDACs), also indicates a potential benefit from HDAC inhibitors (HDACi), such as hydroxamic acid inhibitors (one of which - suberanilohydroxamic acid – is already approved again for treatment of hematological cancers) [92-93]. Inhibitors of HATs (HATi), although not being particularly selective, may also aid in treating patients with SE which show frequently upregulation of these enzymes [94]. The use of such agents might allow for dose reduction of chemotherapy that these young patients endure, as SE is a highly chemo-sensitive solid tumor”.

It would also be helpful to analyze data from other sources such as the GDC data portal.

Reply: We thank the Reviewer for his/her comment. As described in the portal, “The National Cancer Institute’s (NCI’s) Genomic Data Commons (GDC) contains NCI-generated data from some of the largest and most comprehensive cancer genomic datasets, including The Cancer Genome Atlas (TCGA) and Therapeutically Applicable Research to Generate Effective Therapies (TARGET)”. Of the 173 cases of testicular tumors present in the GDC data portal, 151 (87.3%) come from the TCGA project, which was the same database we used in our analysis, with the same cases. Given the type of analyses performed we believe that analyzing the TCGA database through cBioPortal would accomplish the aim of the Review which is to provide clues about most relevant epigenetic biomarkers, obviating the need of downloading raw data from the same samples on GDC.

Round  2

Reviewer 1 Report

The authors improved the manuscript according to my suggestions. 

I accept the manuscript in present form.